# Influence of Rigid Segment Type on Copoly(ether-ester) Properties

**DOI:** 10.3390/ma14164614

**Published:** 2021-08-17

**Authors:** Konrad Walkowiak, Izabela Irska, Agata Zubkiewicz, Zbigniew Rozwadowski, Sandra Paszkiewicz

**Affiliations:** 1Department of Materials Technologies, West Pomeranian University of Technology, Al. Piastów 19, 70-310 Szczecin, Poland; wk42388@zut.edu.pl (K.W.); izabela.irska@zut.edu.pl (I.I.); 2Department of Physics, West Pomeranian University of Technology, Al. Piastów 48, 70-311 Szczecin, Poland; agata.zubkiewicz@zut.edu.pl; 3Department of Inorganic and Analytical Chemistry, West Pomeranian University of Technology, Al. Piastów 42, 71-065 Szczecin, Poland; zbigniew.rozwadowski@zut.edu.pl

**Keywords:** polymer synthesis, polycondensation in melt, polymer modification, 1,6-hexanodiol, dimethyl 2,5-furandicarboxyle, dimethyl terephthalate, dimethyl napthalate, copolymers, biopolymers, polymer characteristics

## Abstract

The growing ecological awareness of society created the tendency to replace petrochemically based materials with alternative energy carriers and renewable raw materials. One of the most requested groups of polymer materials with significant technological importance is thermoplastic elastomers (TPE). They combine the properties of elastomers such as flexibility with the typical properties of thermoplastics, like easy processing. Herein, one compares the influence of rigid segments on the properties of copoly(ester-ether). Thermoplastic polyesters based on bio-1,6-hexanediol and terephthalic (T), furanic (F), and napthalate (N) diesters, i.e., PHT, PHF, and PHN, were obtained employing melt polycondensation. Additionally, to grant elastic properties of polyesters, systems containing 50 wt.% of bio-based polyTHF^®^1000 (pTHF) with a molecular mass of 1000 g/mol, have been prepared. The composition and chemical structure have been determined by ^1^H nuclear magnetic resonance (NMR) and Fourier transformed infrared spectroscopy (FTIR) analyses. The temperatures corresponding to phase transition changes were characterized by differential scanning calorimetry (DSC) and dynamic mechanical thermal analysis (DMTA) analyses. The crystalline structure was examined by X-ray diffraction (XRD) analysis. Additionally, the influence of pTHF–rich segment on the tensile properties, water absorption, as well as thermal and thermo-oxidative stability, has been analyzed. It was found that incorporation of soft phase allows creation of thermoplastic elastomers with tensile characteristics comparable to the commercially available ones, by means of elongation at break higher than 500%, low values of tensile modulus, without exhibiting yield point.

## 1. Introduction

The increasing awareness in the field of ecology within our society has led to the necessity of creating demand for replacing chemical products made from petroleum with renewable raw materials. Therefore, there is a growing interest in materials like poly(tetrahydrofuran) (polyTHF). PolyTHF is a bio-based derivative from 1,4-butanediol [1]. This material possesses characteristic high chain flexibility, and because of that it has a low melting point between −15 °C to 30 °C, which depends on its molecular weight. The bio-based PolyTHF^®^ 1000 produced by BASF has identical properties to a petrochemical-based product [1]. PolyTHF can be used in the production of spandex fibers or artificial leather [2]. Furthermore, it can be used for the synthesis of thermoplastic polyurethane [1,2]. Another material worthy of notice is 1,6-hexanediol (1,6-HDO) produced by Rennovia Inc. Bio-based 1,6-HDO is synthesized from glucose. 1,6-HDO finds applications in the production of coatings, adhesives, and elastomers [3].

Furandicarboxylic acid (FDCA) is one of the most important renewable materials, and it is recognized as one of the most important building-block chemicals by the US Department of energy [4,5]. Thanks to its chemical structure, FDCA is considered an alternative to terephthalic acid (TPA) [4,5,6,7,8]. Therefore, FDCA or its derivatives like dimethyl 2,5-furanodicarboxylate (DMFDC) find use in the synthesis of polyesters [9]. Studies show that poly(ethylene 2,5-furanodicarboxylate) has a higher tensile modulus and barrier performance than PET [4,9]. Thus, PEF can be used in the production of films, bottles, and fibers.

Poly(hexamethylene terephthalate) (PHT) is non-commercial semicrystalline aromatic polyester [8,10,11]. Therefore, PHT has good mechanical properties and chemical resistance [8,11]. PHT is polymorphic polyester, which has three crystal structures: α, β, and γ. The γ is the rarest form because this crystal structure is converted to the other structures by annealing or orientation [12,13]. Replacement of TPA with FDCA, or its derivatives, results in achieving polyesters fully made from renewable materials. In that case poly(hexamethylene 2,5-furanodicarboxylate) (PHF) was synthesized by Moore and Kelly in 1978 [7]. PHF has lower thermal stability and glass transition temperature in comparison to PEF or PTF. The reason for that is the higher methylene number of aliphatic diols in PHF [14]. Same as PHT, PHF is non-commercial material. Polyesters based on 2,6-naphthalene dicarboxylic acid or its derivatives exhibit better or similar properties if compared to polyesters based on TPA. For example poly(ethylene naphthalate) (PEN) has better thermal, mechanical properties, and chemical resistance than PET [15]. The reason for the improvement is a double naphthalene ring in its structure [15]. Poly(hexamethylene naphthalate) (PHN), like PHT, is polymorphic polyester and exhibits two crystallite structures: α and β [12,16]. Similarly, PEN, PHN shows polymorphism induced by temperature and/or stress [17]. PHN investigated by Jeong et al. [16] has melting and glass temperature of 209 and 45 °C, respectively. The crystallization rate of PHN is faster than PEN but slightly slower than that of poly(butylene naphthalate) (PBN) [16]. Thanks to its properties, PHN can be classified as an engineering thermoplastic [16].

The thermoplastic elastomers (TPEs) constitute a group of materials that combine the mechanical properties of elastomers and the processing properties of thermoplastics. Their properties result from morphology, which consists of rigid and flexible segments. A flexible segment forms a matrix in which rigid-segment domains are immersed [18]. Unlike elastomers, the rigid-segment domains of TPE form physical crosslinking [18]. TPE produced by polycondensation or polyaddition can be classified into three groups: polyester-based, polyamide-based, and polyurethane-based [19]. Poly(ether-ester) (PEE) block copolymers are polyester-based TPE, that were commercialized, especially those based on poly(butylene terephthalate) (PBT) as rigid-segment and poly(tetramethylene oxide) (PTMO), also called poly(tetrahydrofuran) (polyTHF) as flexible segment. The most popular commercial representative of these materials is Hytrel^®^ (DuPont) [20]. TPEs are widely used in many commercial applications, like for the production of bottles, films, caps, etc. TPEs also find applications in medical, surgical, and pharmaceutical activities [19]. However, TPEs are most widely used in the automotive industry, in the production of mounting elements, vibration control elements, floor mats, cup holders, and cable sleeves [19].

This study aims to analyze the influence of rigid segment type on the properties of copoly(ether-ester)s. The PHF, PHT, PHN, and copoly(ether-ester)s by means of PHF-b-F-pTHF 50/50, PHT-b-T-pTHF 50/50, and PHN-b-N-pTHF 50/50 were synthesized via melt polycondensation. The composition and chemical structure were determined by ^1^H nuclear magnetic resonance (NMR) and Fourier transform infrared spectroscopy (FTIR). The temperatures corresponding to phase transition changes were investigated by differential scanning calorimetry (DSC) and dynamic mechanical thermal analysis (DMTA). The crystalline structure of the synthesized materials has been evaluated by X-ray diffraction (XRD). The influence of flexible segment on static tensile properties, intrinsic viscosity, and water absorption was evaluated. Besides, thermal and thermo-oxidative stability has been analyzed, to evaluate the utilitarian properties of the synthesized polyesters and copoly(ether-esters).

## 2. Materials and Methods

### 2.1. Synthesis of PHF, PHT, PHN, and Copoly(ether-ester)s Based on Them

Synthesized polyesters and PEE were obtained from dimethyl terephthalate (DMT, Sigma-Aldrich, Germany), dimethyl naphthalene (DMN, Sigma-Aldrich, Germany), dimethyl 2,5-furanodicarboxylate (DMFDC, 99%, Henan Coreychem Co., Ltd., Zhengzhou, China), 1,6-hexylene glycol (HDO, Rennovia Inc., Santa Clara, CA, USA), and poly(tetrahydrofuran) (pTHF, BASF, Ludwigshafen, Germany) with a molecular mass of 1000 g/mol. Materials were synthesized in two steps (Figure 1). In the first step, transesterification of diester (DMFDC/DMT/DMN) by HDO with the presence of the first portion of catalyst (tetrabuthyl orthotitanate, Ti(OBu)_4_, (Fluka)) was carried out. Secondly, the polycondensation stage was performed in the presence of the second portion of catalyst (also Ti(Obu)_4_) along with the thermal stabilizer, Irganox 1010 (Ciba-Geigy, Basel, Switzerland). Additionally, during polycondensation of copoly(ether-ester)s pTHF was added. The synthesis was performed in a 1 dm^3^ high-pressure reactor (Autoclave Engineers, Erie, PA, USA). The reactor is equipped with a condenser, cold trap for collecting the by-product, and vacuum pump. During the transesterification, the reactor was loaded with DMFDC/DMT/DMN, HDO, and catalyst. The molar ratio of the diester (DMFDC/DMT/DMN) and glycol (HDO) was 1:1.5. The first step of reaction was performed in the presence of a constant flow of nitrogen at the temperature and time that can be found in Table 1. During the transesterification, methanol was distilled and collected as the first by-product. The end of the first step was signalized by the amount of effluent by-product. When 90% of the stoichiometric amount of methanol was ceased, the process was completed. Subsequently, one increased the temperature up to 210 °C and added a catalyst and thermal stabilizer to the reactor. Additionally, if it was the synthesis of copoly(ether-ester)s, pTHF was added to the reactor. The reaction temperature was increased to a value of the end temperature of the polycondensation (Table 1). A vacuum was applied to remove HDO excess and lower the final pressure to 25 Pa. The progress of polycondensation was monitored by observation of stirring torque change, which was used to evaluate the melt viscosity of the product. The end of the polycondensation process was signalized by the proper value of melt viscosity of the reaction mixture that was adequate to the value of melt viscosity of a high molecular mass of the polymer material. After the polycondensation process, the material was extruded from the reactor into the water bath with the use of compressed nitrogen. All synthesized copoly(ether-ester)s were named as follows: rigid segment-b-diester unit—flexible segment, since for the flexible segment one diester unit (F/T/N) is included with pTHF sequence, e.g., PHF-b-F-pTHF. In the same manner, all of the other copoly(ether-esters) based on diester of terephthalate (T) and naphthalene (N) acids were designated.

Subsequently, the extruded materials were pelletized on the laboratory mill. Then materials were injection molded using Boy 15 (Dr Boy GmbH&Co., Neutstadt, Germany) for a dumbbell shape sample, type A3, that was being used for DMTA, water absorption, and tensile measurements. Before injection molding, pellets were dried for 24 h under a vacuum at the temperature of 55 °C. The parameters of injection can be found in Table 2.

### 2.2. Characterization Methods

The chemical structure and composition of the synthesized materials were investigated by ^1^H NMR spectroscopy. Before the experiment, samples were subjected to esterification. ^1^H NMR spectroscopy was carried out on Bruker spectrometer operating at a frequency of 400 MHz (Bruker, Karlsruhe, Germany). The chloroform-d CDCl3 at a concentration of 10 mg/mL was used to dissolve samples. For internal chemical shift reference, tetramethylsilane (TMS) was used.

The polyesters and copoly(ether-ester)s were characterized using an FTIR spectrophotometer Nicolet iS 5 FTIR Spectrometer (Thermo Fisher Scientific Inc., Waltham, MA, USA). Samples were tested with the attenuated total reflectance (ATR) technique. Each sample was scanned 17 times over the frequency range of 4000–400 cm^−1^.

The intrinsic viscosity (IV) of synthesized polyesters and copoly(ether-ester)s was investigated at 30 °C in the mixture of phenol/1,1,2,2- tetrachloroethane (60/40 by weight). The concentration of polymer solution was 0.5 g/dl. The intrinsic viscosity was characterized using a capillary Ubbehlode viscometer (type Ic, K = 0.03294).

Density was studied by hydrostatic weighing method with the use of AGN200C (Axis LLC, Gdańsk, Poland).

Additionally, water absorption tests were performed in cold and boiling water. The measurement was carried out according to the procedures recommended in ASTM D570. Firstly, the dumbbell shape samples were dried at 55 °C for 24 h. Subsequently, samples were cooled to room temperature and weighted. The measurement of water absorption in boiling water was carried out for 30 min, after that samples were cooled in distilled water for 15 min. The water absorption in cold water was investigated by immersing samples in distilled water at 23 °C for 24 h. The water on the surface of the samples was removed with filled paper. After that, samples were weighted. Each reported value is an average of 3 test specimens.

Differential Scanning Calorimetry (DSC) measurement was performed using F1 Phoenix (Netzsch, Selb, Germany). The measurements were carried out over a temperature range of −50 °C to 300 °C and again cooled to −50 °C with a speed of 10 °C/min for polyesters. Copoly(ether-ester)s were investigated in the temperature range of −90 °C to 300 °C and again cooled to −90 °C with the same speed as polyesters. Additionally, only PHT and PHT-b-T-pTHF were heated to 250 °C. First heating was used only for deleting the thermal memory of the samples. In addition, in order to compare the DSC results with dynamic mechanical thermal analysis (DMTA) results, by means of phase transition temperatures, the DSC measurement was also carried out with the heating/cooling rates of 3 °C/min in the temperature ranges adequate for polyesters and copoly(ether-esters), as above. The melting and crystallization temperatures (T_m_, T_c_) were measured at the maximum of endo- and exothermic peaks, respectively. The glass transition temperature (T_g_) was taken as the midpoint of the change of the heat capacity (ΔC_p_).

The X-ray diffraction (XRD) patterns of the catalysts were recorded with Empyrean (Malvern Panalytical, Malvern, UK) using CuK (λ = 0.154 nm) as the radiation source. The samples were scanned in a 2θ angle range of 10–35° (with a step size of 0.05).

The dynamic mechanical thermal analysis (DMTA) was obtained using a DMA Q800 (TA Instruments, New Castle, DE, USA) working in a temperature range from −100 °C to the polymer melt temperature, at a frequency of 1 Hz and a heating rate of 3 °C/min. The properties were determined based on modulus changes and the ability of attenuation as a function of temperature and frequency of load changes.

The thermal and thermo-oxidative stability of materials were characterized using TGA92-16.18 (Setaram, Caluire-et-Cuire, France). The thermal stability was investigated in the atmosphere of argon when thermo-oxidative in the atmosphere of dry, synthetic air (N_2_:O_2_ = 80:20% vol) with a flux rate of 20 mL/min. The study was carried out in the temperature range of 50–700 °C at a heating rate of 10 °C. The static mechanical properties were investigated using Autograph AG-X plus (Shimadzu, Kyoto, Japan). This machine is equipped with an optical extensometer, 1 kN Shimadzu load cell, the TRAPEZIUM X software. Polyesters and copoly(ether-ester)s at the beginning were extended to 1% with a crosshead speed of 1 mm/min. Then, the stress-strain curves for polyesters were obtained at a rate of 5 mm/min. However, the stress-strain curves for copoly(ether-ester)s were obtained at a rate of 100 mm/min. Each reported value is an average of five test specimens. The measurements were performed according to PN-EN ISO 527. Finally, the hardness of polyesters and copoly(ether-ester)s was investigated using a Zwick 3100 Shore D tester (Zwick GmbH, Ulm, Germany). The reported values are the mean values of twenty independent measurements.

## 3. Results

### 3.1. Chemical Structure, Composition, and Basic Physico-chemical Properties

Polyesters (PHF, PHT, PHN) and copoly(ether-ester)s based on these polyesters were synthesized by melt polycondensation which was described in the experimental session.

The theoretical chemical composition and the chemical composition estimated from ^1^H NMR analysis are summarized in Table 3. The NMR spectra of polyesters shown in Figure 2 differ from each other. For PHF, two furanoate ring protons appear as a singlet signal at 7.20 ppm (d signal) [7]. The peak at 8.07 ppm (e signal) is assigned to the protons of the terephthalate ring. Signals at 8.59, 8.07, and 7.98 ppm (f, g, h signals) on the ^1^H NMR spectra of PHN can be ascribed to protons of naphthalene ring [21]. The signals originating from methylene protons of hexylene glycol units appear at 1.45, 1.77, and 4.33 ppm for PHF [22]. However, signals of methylene protons of hexylene glycol units slightly differ for other polyesters (PHT, PHN). They can be found at higher values due to the impact of the aromatic ring on a magnetic field.

Figure 3, Figure 4 and Figure 5 show a comparison of ^1^H NMR spectra of polyesters and block copolymers. The ^1^H NMR spectra of poly(ether-ester)s have four new resonance signals assigned to the protons of pTHF. Signal i is assigned to inner methylene protons of pTHF ether block at 1.6 ppm and outer methylene protons of ether block (signal k) that can be found at 3.42 ppm [22,23]. The last two new signals are caused by a transesterification reaction between polyesters and pTHF what leads to a change in resonance of methylene group proton in pTHF. This effect is visible at 4.37 and 1.64–1.73 ppm (signal l and j, respectively). Moreover, due to the impact of the naphthalene ring on magnetic field peaks ascribed to signals a, i, and c, l have merged. Signal m is visible in ^1^H NMR spectra of PHN-b-N-pTHF 50/50 at 3.47–3.52 ppm and is ascribed to protons from the ending group of macromolecule -CH_2_-OH [24].

For estimation of the actual weight content of flexible segment, the Equation (1) was used:(1)$$W_{F}\left(\%\right)=\frac{\left(\frac{I_{F}}{y}\right)\times M_{\mathrm{WF}}}{\left(\frac{I_{F}}{y}\right)\times M_{\mathrm{WF}}+\left(\frac{I_{R}}{x}\right)\times {\;M}_{\mathrm{WP}}\;}$$
where I_R_ is the internal signal intensities corresponding to rigid segment I_F_ is the internal signal intensities corresponding to flexible segment, x is a number of protons assigned for an internal signal of a rigid segment (for all block copolymers x = 4), y is a number of protons assigned for an internal signal of the flexible segment (for all block copolymers y = 6), M_WP_ and M_WF_ are the molecular weights of the repeating units of the rigid and flexible segment, respectively.

The difference between calculated composition and expected values is around 5%. The most accurate copoly(ether-ester) is PHF-b-F-pTHF 50/50, which calculated composition differs only about 2,49% from the expected value. The difference can be caused by the introduction of additional pTHF, due to losses in the dosing of the viscous substrate and distillation of the reaction substrate under reduced pressure [22].

The FTIR analysis of polyesters and copoly(ether-ester)s was carried out to investigate their structure. In Figure 6, FTIR spectra of polyesters and copoly(ether-ester)s shown similarities that occur between 1255–1270 cm^−1^ and 1708–1715 cm^−1^. Absorption peaks in that area are caused by C(=O)-O and C=O stretching mode of ester groups, respectively [7,21]. There are also characteristic bands of aromatic ring moieties shown near 724–764 cm^−1^ (ring out-of-plane deformation), 820–875 cm^−1^ (ring out-of-plane deformation), 1016–1130 cm^−1^ (ring in-plane deformation) [25,26]. Moreover, for PHF and PHF-b-F-pTHF signal associated at 1222 cm^−1^, arising from the =C-O-C= furan ring vibration [22].

The influence of soft segment on FTIR spectra is visible, especially at characteristic bands of C-O-C stretching vibration of pTHF between 1105–1125 cm^−1^. Consequently, after adding pTHF there is a visible increase in absorption at a wavelength of 2847–2852 cm^−1^ and 2915–2919 cm^−1^ due to -C-H asymmetric and symmetric stretching vibrations in methylene groups [27]. Additionally, for PHF-b-F-pTHF 50/50 and PHT-b-T-pTHF 50/50 the absorption peak, associated with C=O stretching mode of the ester group, slightly increased. Finally, the absorption band of -OH groups near 3450 cm^−1^ is very weak. Therefore, it indicates the consumption of pTHF hydroxyl groups through copolymerization, which confirms the successful modification of polyesters [22,24].

The above observations on the chemical structure and composition of polyesters and copoly(ether-esters) based on 1,6-hexanediol are in the agreement with the previously published papers [10,12,22].

In addition, the polyesters and copoly(ether-ester)s were investigated in terms of limiting viscosity number, density, and water absorption. The results are summarized in Table 3. The polyester with the highest limiting viscosity number is PHT, moreover, PHN exhibits the lowest value of limiting viscosity number (0.626 dl/g) among other polyesters. Nevertheless, all synthesized polyesters exhibited high values of intrinsic viscosity, especially if compared to PET bottle grade (IV of: 0.70–0.78 (dl/g) [28], or to the previously synthesized PET (IV if 0.536 dl/g, Mn = 19,500 g/mol and Mw = 46,900g/mol [29]. Whereas, the previously synthesized furan-based polyesters [30] and copoly(ether-esters) [22], following the same procedure, also exhibited relatively high values of molecular masses. Obviously, the incorporation of flexible segments into polyester caused the increase in the value of intrinsic viscosity, which is due to change in molecular chains masses and their flexibility. This is in the agreement with our previously published papers [20,22]. All synthesized polyesters exhibited similar values of hydrostatic density. Whereas, the addition of pTHF lowers values of density of copoly(ether-ester)s. Figure 7 represents the water absorption of synthesized polymers at room temperature (CWA at 23 °C) and boiling water (HWA at 100 °C). It is well-known that polyesters absorb water, which affects their processing properties, thus of great importance is to evaluate their affinity for water absorption [30]. Polymer materials absorb water and moisture by various mechanisms [9]. One of them is due to the diffusion of water molecules to free volume between polymer chains [9,29,30]. PHN has the lowest value of absorption in both, cold and hot water. PHF and PHT exhibited similar values of cold water absorption. However, PHF shows a higher absorption of hot water than PHT. In turn, copoly(ether-ester)s show significantly higher values of cold water absorption than polyesters. The lower values of HWA in the case of block copolymers might be due to the relaxation in the temperature much above the glass transition.

### 3.2. Structural Properties

For an investigation of structural and thermal properties, DSC, DMTA, and XRD were used. DSC analysis is summarized in Table 4. Figure 8 represents DSC thermograms of synthesized polyesters recorded during cooling and second heating at heating/cooling rates of 10 °C/min and 3 °C/min. Polyester with the highest value of glass transition temperature is PHN, which was due to the double aromatic ring in its chain, lowering its mobility. The glass transition temperatures of PHF and PHT are similar to one another. PHN has the highest value of melting temperature and it’s the only synthesized polyester that shows cold crystallization at 194.7 °C at a heat rate of 10 °C/min. Whilst PHF has the lowest value of melting temperature. In turn, PHT has two melting temperatures, resulting from two crystal structures: α and β [13]. Moreover, all investigated polyesters were found to be semi-crystalline polymers, wherein the crystallization temperatures at heat rate of 10 °C/min of PHF (T_c_ = 96.8 °C) and PHT (T_c_ = 105.9 °C) appear to be very similar, while PHN has the highest value of crystallization temperature of 169.0 °C.

Figure 9 represents DSC thermograms of polyesters and copoly(ether-ester)s at heating/cooling rates of 10 °C/min and 3 °C/min. One can clearly observe the influence of flexible segment on the phase transition temperatures (T_c_, and T_m_) that were shifted toward lower values. The phase separation between rigid and flexible segments was not visible at a heat rate of 10 °C/min due to crystallization of rigid segment during cooling, although one can observe two glass transitions at a heat ratio of 3 °C/min. For PHF-b-F-pTHF 50/50 and PHT-b-T-pTHF 50/50 value of glass transition of the rigid segment increased compared to PHF and PHT, respectively. In turn, PHN-b-N-pTHF 50/50 has a lower value of glass transition of the rigid segment in comparison to the corresponding polyester. 

The crystal structure of the synthesized materials was investigated using XRD analysis. Figure 10 represents diffraction patterns of synthesized polyesters and copoly(ether-ester)s. PHF’s WAXS diffractogram has three sharp signals at 2θ = 13.67°, 17.03°, 24.91° which are assigned respectively to planes (110), (010), (111) [14,22]. Similar to PHF, in WAXS diffractogram of PHN, three sharp signals at 2θ = 16.40°, 21.34°, 23.64° that are assigned respectively to planes (010), (11$\bar{1}$), (100) can be observed [31]. In the case of PHT, the polymorphism, suspected from DSC analysis, was confirmed. The α structure signals can be observed at 2θ = 16.29°, 19.55°, 20.70°, 21.44°, 25.54° [13] and signals that are coming from β structure can be observed at 2θ = 18.18°, 23.64° [13]. The WAXS diffractogram proves different crystal structures of all synthesized polyesters. In all cases of copolymers, peaks on the WAXS curves are less sharp than for polyesters due to a lower degree of crystallinity (estimated by DSC). It is especially noticeable for the β structure of PHT-b-T-pTHF 50/50 where that structure is almost not visible in the diffraction pattern.

Figure 11 represents the results of DMTA where the storage modulus (E’), and tan δ are plotted as a function of temperature at 1 Hz. The values of E’ at 25 °C and α-relaxations are summarized in Table 5. The sample with the highest value of E’ at 25 °C among polyesters is PHF (2577 MPa), while PHT has the lowest value (2105 MPa). Copolymers have significantly lower values of the storage modulus at 25 °C. In the group of copoly(ether-ester)s PHF-b-F-pTHF 50/50 exhibits the value of E’ at 25 °C higher than the others (181 MPa) when PHT-b-T-pTHF 50/50 and PHN-b-N-pTHF 50/50 differ only about 1 MPa. Copolymers show a wide plateau of elasticity that determines the possibility of using them as thermoplastic elastomers. Figure 11a represents tan δ for polyesters, where α-relaxation what can be recognized as T_g_ is observed. PHN has the highest value of α_1_ (73 °C) and PHT has the lowest value of α_1_ (43 °C). For copoly(ether-ester)s one can clearly observe two peaks at tan δ, which are associated with α-relaxations of rigid (α_1_) and flexible (α_2_) segments. Temperatures of α-relaxations compared to values of T_g_ resulting from DSC analysis at a heat rate of 3 °C/min exhibit similarities in case of values α_1_ and T_gR_. The values of α_1_ for PHF-b-F-pTHF 50/50 and PHT-b-T-pTHF 50/50 are higher than for PHF and PHT, respectively, when PHN-b-N-pTHF 50/50 has a lower value of α_1_ in comparison to PHN. Moreover, PHF-b-F-pTHF 50/50 has the highest value of α_1_-relaxation between copoly(ether-ester)s. In turn, PHN-b-N-pTHF 50/50 has the lowest value of α_1_-relaxation, although it has the highest value of α_2_-relaxation what can indicate partial miscibility of amorphous phases of PHN and pTHF. 

### 3.3. Thermal Properties

The thermal and thermo-oxidative stability of synthesized materials was investigated due to the importance of these parameters for the future application of materials. The thermal stability of materials was performed under an inert atmosphere (argon) and thermo-oxidative stability was carried out under oxidative atmosphere (air). The characteristic temperatures for the mass losses of 5, 10, 50, 90% (T_5%_, T_10%_, T_50%_, T_90%_) in both atmospheres and the temperatures corresponding to the maximum of mass losses (T_DTG1_ and T_DTG2_) are summarized in Table 6. In this study, the temperature for the mass loss of 5% is considered as the beginning of thermal degradation. Figure 12a,e represent the thermal-oxidative and thermal stability of polyesters, respectively. The values of each characteristic temperature for the mass losses and the temperatures corresponding to the maximum of mass losses in both atmospheres are almost similar. PHN has the best thermal and thermo-oxidative stability among investigated polyesters. The polyester which is most sensitive for degradation is PHF. The difference between values of T_5%_, and T_DTG1_ in both atmospheres for PHF and PHN is almost about 20 °C.

Figure 12b–d,f–h represents thermo-oxidative and thermal stability of polyesters and copoly(ether-ester)s, respectively. The influence of soft segment in copoly(ether-ester)s is significant. Especially, it is noticeable in the decrease in the value of T_5%_ of mass loss of copoly(ether-ester)s in comparison to corresponding polyesters. The first derivative of the mass loss (DTG) curve in Figure 12b–d shows two-step decomposition under an oxidative atmosphere. The first step appears at 386–405 °C and its attributed to the decomposition of flexible and rigid segments. The second step appears at 470–529 °C, and it results from the decomposition of residue. The values of T_DTG1_ for polyesters and copoly(ether-ester)s do not show great differences, only about 2 °C. Although, there is a significant difference in the value of T_DTG2_ between them. For PHF-b-F-pTHF 50/50 value of T_DTG2_ is 24 °C higher than for PHF. A similar effect was observed by Szymczyk et al. [32] for poly(trimethylene terephthalate)-*block*-poly(ethylene oxide) (PTT-b-PEO) copolymers. The value of T_DTG2_ for PHT and PHT-b-T-pTHF 50/50 is almost similar; furthermore, PHN has a higher value of T_DTG2_ than PHN-b-N-pTHF 50/50.

### 3.4. Mechanical Properties

Figure 13 represents the stress-strain curves of the synthesized materials. The values of Young modulus (E), tensile strength at yield (σ_y_), elongation at yield (ε_y_), tensile strength at break (σ_b_), and elongation at yield (ε_y_) are summarized in Table 7. All polyesters show a strain hardening effect due to the orientation of macromolecular chains and crystallization during stretching. PHN has the highest value of Young modulus among other polyesters, by about 1 TPa. Moreover, the values of tensile strength at yield and tensile strength at break of PHN are higher than for PHF and PHT. However, the polyester that characterizes significantly better elongation is PHT, which elongation at break differs from others around 100%. The addition of soft segment results in the disappearance of yield point, significant increase value of elongation at break, and decrease in value of tensile strength at break. The disappearance of yield point is due to the initiation of highly elastic deformations at low deformation. These characteristics are typical for elastomeric materials. The most promising copoly(ether-ester) is PHN-b-N-pTHF 50/50. This material has the highest value of tensile strength at break and its value of elongation at break classify between PHF-b-F-pTHF 50/50 and PHT-b-T-pTHF 50/50. Due to the addition of pTHF ability to crystallization decreased and it caused a decrease in a strain hardening effect. The hardness of polyesters and copoly(ether-ester)s are summarized in Table 7. The polyester with the highest value of Shore D hardness in PHN. The hardness value of PHF and PHT slightly differs. However, the addition of soft segment results in a decrease in hardness in copoly(ether-ester)s in comparison to polyesters.

## 4. Conclusions

The synthesis of PHF, PHT, PHN, and copoly(ether-ester)s by two-step melt polycondensation was carried out successfully. Synthesized materials were investigated and compared in terms of chemical structure, mechanical, thermal, utilitarian properties. The chemical structure of materials was analyzed by FTIR spectroscopy and ^1^H quantitative nuclear magnetic resonance (NMR) and it confirmed that real composition is similar to feed one. Using DSC, DMTA, TGA thermal properties and phase structure of synthesized materials were studied. The crystal structure was investigated by XRD analysis. The static tensile performance was carried out for polyesters and copoly(ether-ester)s. Moreover, intrinsic viscosity number, density, and water absorption were studied. All investigated polyesters are semicrystalline. The obtained copolymers are characterized by high values of deformation. Additionally, synthesized copolymers can be processed by injection molding. These characteristics prove the attainment of TPE. Among synthesized materials, the most prominent are PHN and PHN-b-N-pTHF 50/50 due to their properties. They characterize by the most rigid macromolecular chain since their value of glass transition temperature and thermal stability are significantly higher than others. Moreover, their values of hardness, Young modulus, and tensile strength at break are higher as well. However, dimethyl naphthalate is more expensive than traditional TPA so complete replacement of it is not profitable. PHF and PHT show similarities in phase transition temperatures and hardness. PHT has slightly better thermal and thermal-oxidative properties than PHF, although PHF is characterized by a higher value of Young modulus and tensile strength at break than PHT. In the future, PHF could be prioritized over PHT due to being a bio-based material and a decrease in the price of FDCA. Moreover, if comparing the synthesized copoly(ether-ester)s, based on PHF and PHN, with commercially available thermoplastic elastomers with similar values of hardness (Shore hardness ca. 40 ShD)—for instance, Hytrel^®^ 4056 [33,34]—the obtained materials are found to have higher values of elongation and tensile stress at break. A slightly lower value of σ_b_ was observed for PHT- b-T-pTHF 50/50, when keeping the excellent value of ε_b_, comparable to the values typically observed for commercially available TPE. The above-mentioned statement clearly demonstrates the application potential of the synthesized TPEs based on all employed diesters.

## Figures and Tables

**Figure 1 materials-14-04614-f001:**
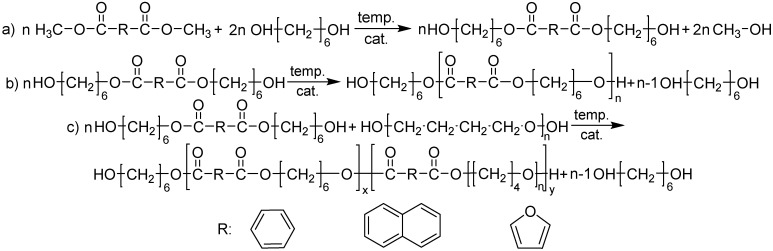
Synthesis reaction of (**a**) transesterification of polyesters and copoly(ether-ester)s; (**b**) polycondensation of polyesters; (**c**) polycondensation of copoly(ether-ester)s.

**Figure 2 materials-14-04614-f002:**
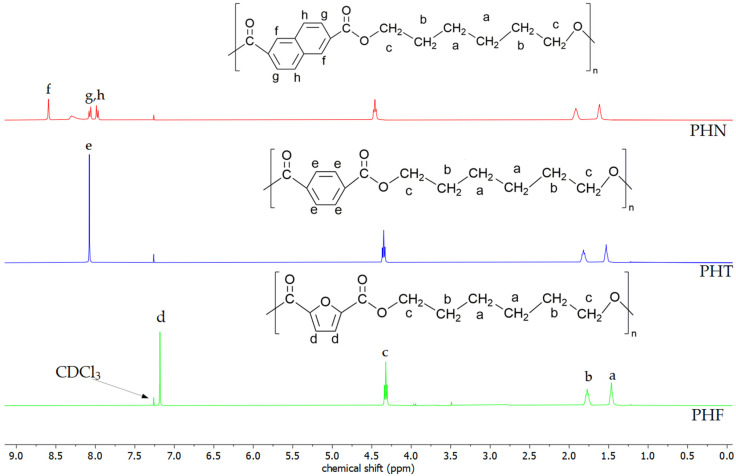
Structure of polyesters and their NMR spectra.

**Figure 3 materials-14-04614-f003:**
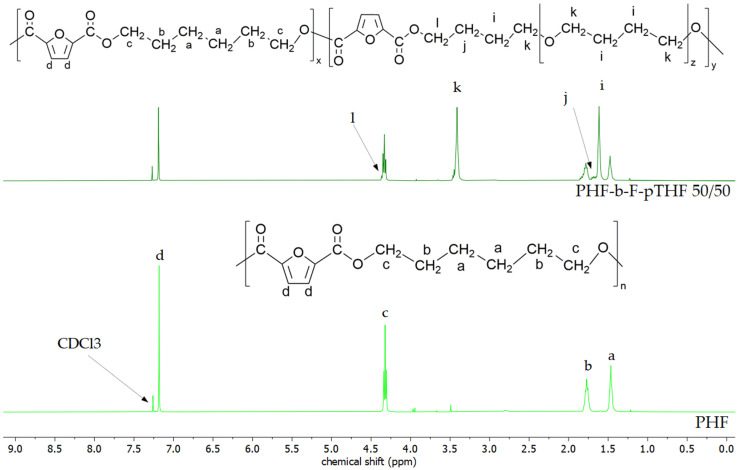
Structure and NMR spectra of PHF and PHF-b-F-pTHF 50/50.

**Figure 4 materials-14-04614-f004:**
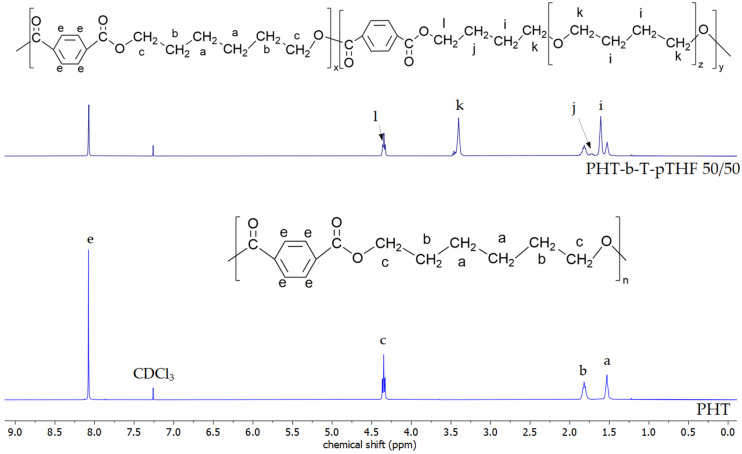
Structure and NMR spectra of PHT and PHT-b-T-pTHF 50/50.

**Figure 5 materials-14-04614-f005:**
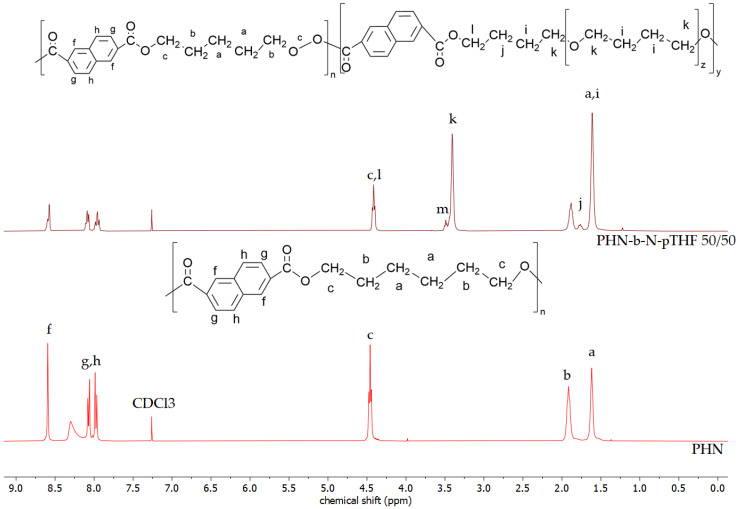
Structure and NMR spectra of PHN and PHN-b-N-pTHF 50/50.

**Figure 6 materials-14-04614-f006:**
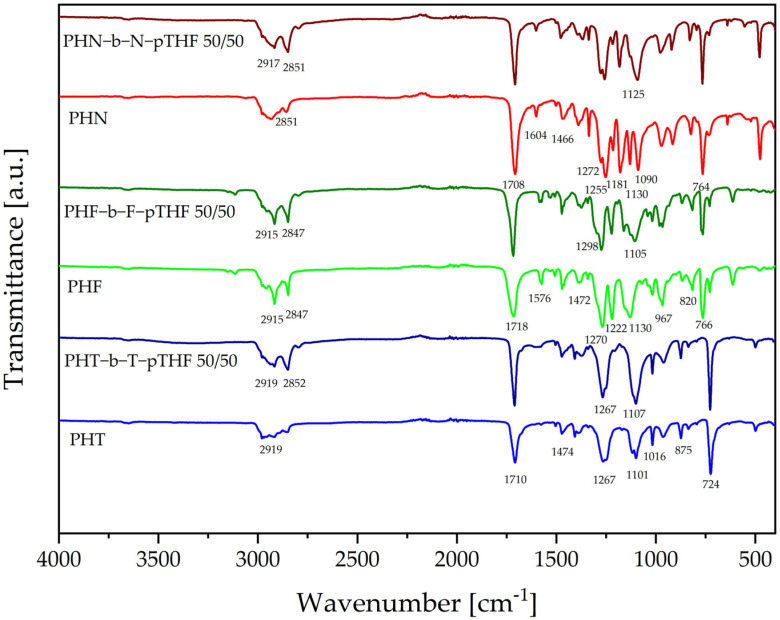
FTIR spectra of the synthesized materials.

**Figure 7 materials-14-04614-f007:**
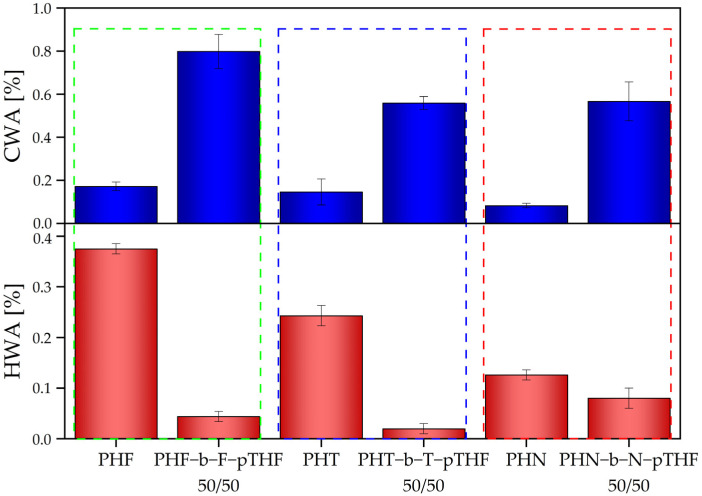
Water absorption in the hot and cold water of PHF, PHT, PHN, PHF-b-F-pTHF 50/50, PHT-b-T-pTHF 50/50, PHN-b-N-pTHF 50/50.

**Figure 8 materials-14-04614-f008:**
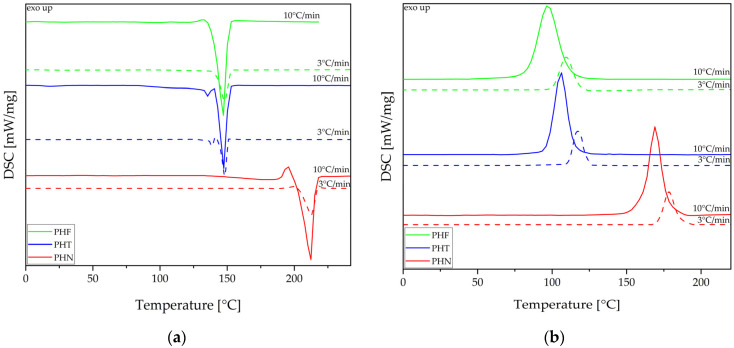
Differential scanning calorimetry (DSC) thermograms were recorded during the second heating (**a**) and cooling (**b**) of polyesters.

**Figure 9 materials-14-04614-f009:**
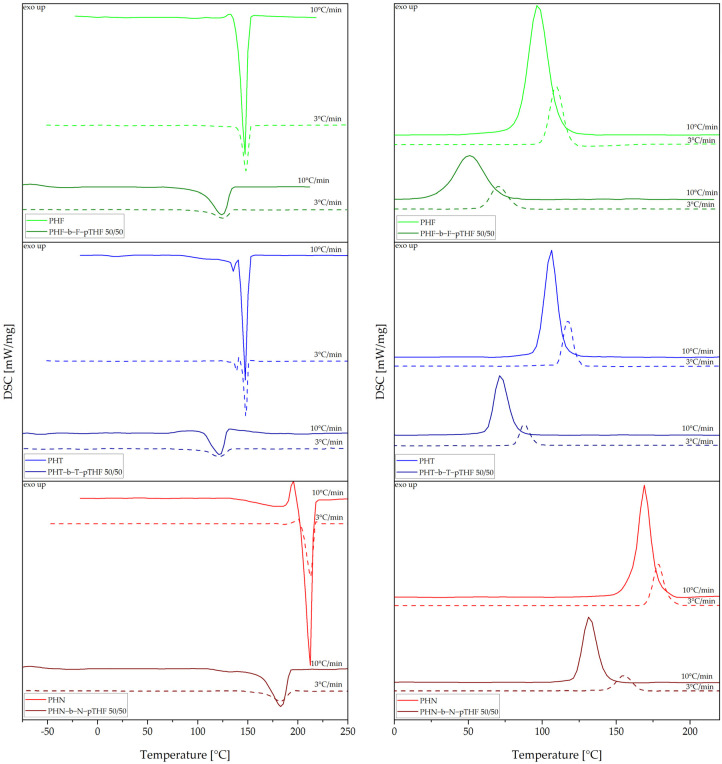
DSC thermograms were recorded during second heating (**left** panel) and cooling (**right** panel) for polyesters and copoly(ether-ester)s.

**Figure 10 materials-14-04614-f010:**
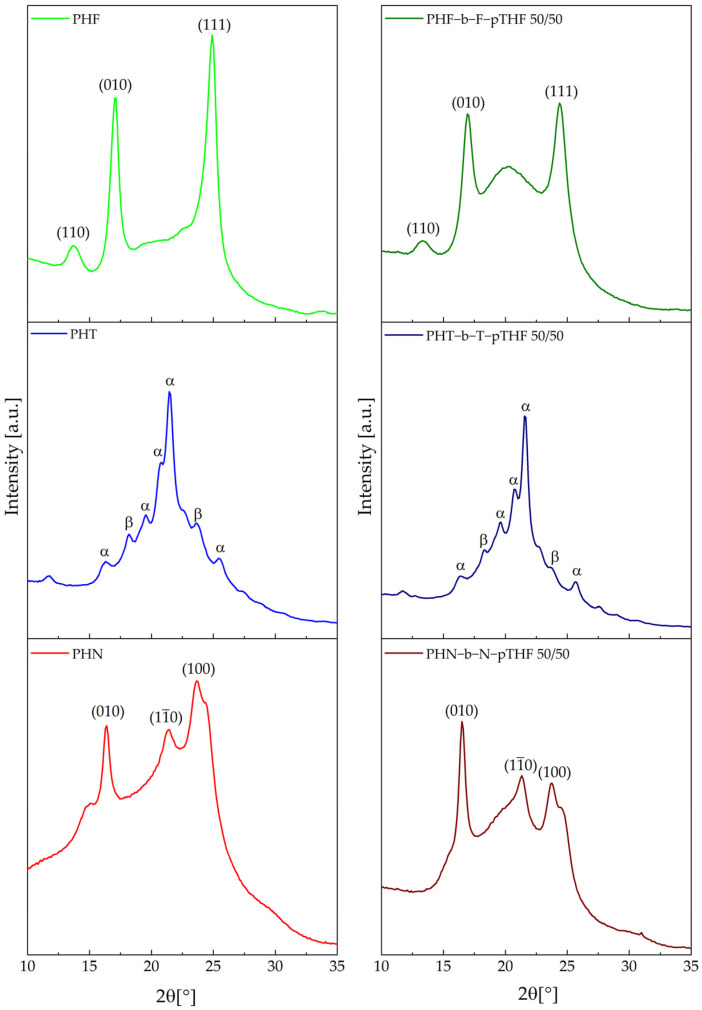
X-ray diffraction (XRD) curves of synthesized polyester and copolymers.

**Figure 11 materials-14-04614-f011:**
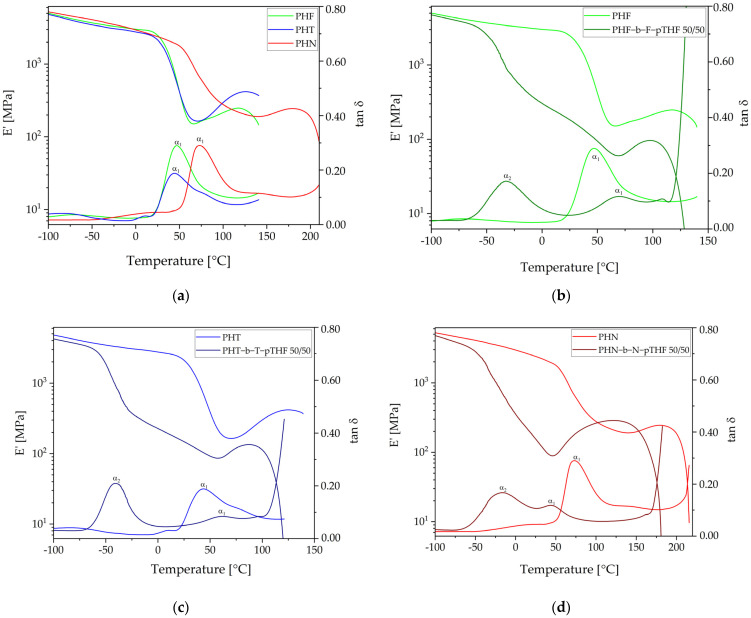
The storage modulus E’ (**a**,**c**) and tan δ (**b**,**d**) as a function of temperature for polyesters and copoly(ether-ester)s.

**Figure 12 materials-14-04614-f012:**
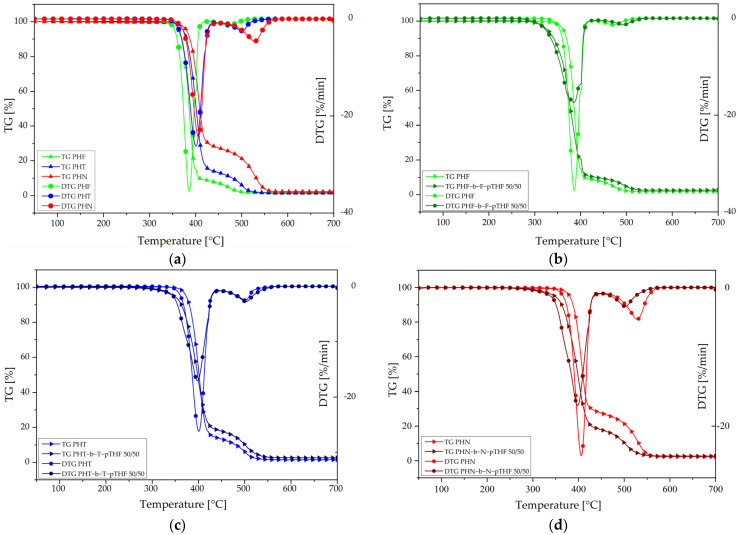
TG and DTG curves for synthesized materials measured in an oxidizing atmosphere (**a**–**d**) and in an inert atmosphere (argon) (**e**–**h**) at the heating rate of 10 °C/min.

**Figure 13 materials-14-04614-f013:**
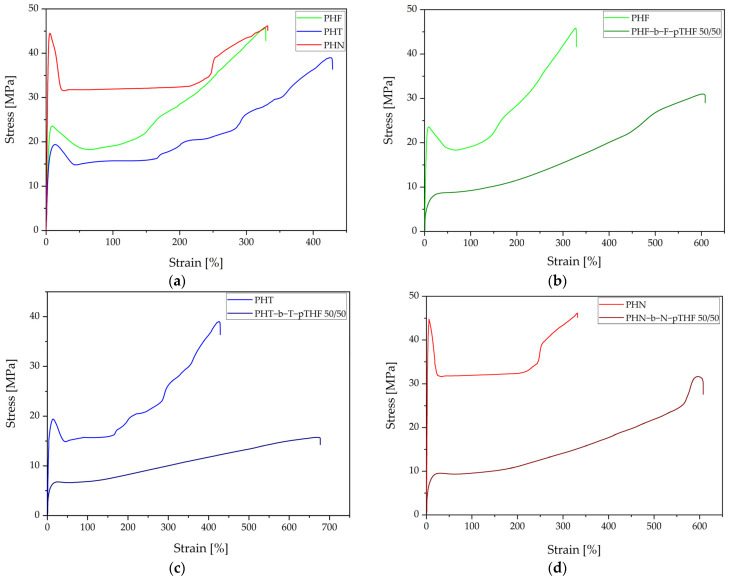
Representative stress-strain curves of static tensile test for synthesized materials (**a**) polyesters, (**b**) PHF and PHF-b-F-pTHF 50/50, (**c**) PHT and PHT-b-T-pTHF 50/50, (**d**) PHN and PHN-b-N-pTHF 50/50.

**Table 1 materials-14-04614-t001:** Temperatures and times used in the synthesis of polyesters and copoly(ether-ester)s.

Sample	Transesterification			Polycondensation	
	T_p_ (°C)	T_kT_ (°C)	Time (min)	T_kP_ (°C)	Time (min)
PHF	165	185	70	235	370
PHF-b-F-pTHF 50/50	165	185	96	235	344
PHT	175	185	85	255	350
PHT-b-T-pTHF 50/50	175	185	93	250	423
PHN	175	195	125	255	285
PHN-b-N-pTHF 50/50	175	190	125	250	388

T_p_—starting temperature of transesterification; T_kT_—ending temperature of transesterification; T_kP_—ending temperature of polycondensation.

**Table 2 materials-14-04614-t002:** Injection molding parameters of polyesters and copoly(ether-ester)s.

Sample	T_f_ (°C)	T_i_ (°C)	P_d_ (MPa)	P_i_ (MPa)	t_i_ (s)	t_c_ (s)
PHF	30	180	20	55	7	60
PHF-b-F-pTHF 50/50	30	170	8	45	7	25
PHT	30	190	30	70	5	20
PHT-b-T-pTHF 50/50	30	150	12	50	6	6
PHN	30	230	25	50	5	10
PHN-b-N-pTHF 50/50	30	210	25	50	5	5

T_f_—temperature of form; T_i_—temperature of injection; P_d_—holding down pressure; P_i_—injection pressure; t_i_—time of injection; t_c_—time of cooling.

**Table 3 materials-14-04614-t003:** Calculated composition and basic physico-chemical properties of synthesized polymers.

Material	M_R_	M_F_	W_R_	W_F_	[η] (dl/g)	d (g/cm^3^)
PHF	-	-	-	-	0.825	1.260
PHF-b-F-pTHF 50/50	238.24	210	47.51	52.49	1.350	1.137
PHT	-	-	-	-	0.952	1.235
PHT-b-T-pTHF 50/50	248.28	220.22	47.09	52.91	1.026	1.132
PHN	-	-	-	-	0.626	1.247
PHN-b-N-pTHF 50/50	298.34	270.29	43.62	56.38	1.176	1.152

M_R_—molecular mass of rigid segment; M_F_—molecular mass of flexible segment; W_R_—calculated content of rigid segment; W_F_—calculated content of rigid segment; [η]—intrinsic viscosity; d—density.

**Table 4 materials-14-04614-t004:** Thermal properties determined from cooling and 2nd heating thermograms for synthesized polyesters and copolymers.

Sample	T_gR_ [°C]	ΔC_p_ [J/g °C]	T_gF_ [°C]	T_c_ [°C]	ΔH_c_ [J/g]	T_m_ [°C]		ΔH_m_ [J/g]	
PHF	14.7	0.12	-	96.8	51.3	147.0		49.8	Thermal properties at heating/cooling rate 10 °C/min
PHF-b-F-pTHF 50/50	-	0.25	−55.7	51.5	27.5	123.6		23.0	
PHT	13.3	0.11	-	105.9	38.6	136.4	147.6	64.5	
PHT-b-T-pTHF 50/50	-	0.12	−67.3	71.7	23.3	121.5		23.4	
PHN	54.0	0.12	-	169.0	46.8	212.0		44.1	
PHN-b-N-pTHF 50/50	-	0.20	−52.2	132.0	27.0	183.5		35.2	
PHF	19.8	0.12	-	109.4	55.3	148.1		55.6	Thermal properties at heating/cooling rate 3 °C/min
PHF-b-F-pTHF 50/50	55.8	0.31	−55.2	70.4	27.9	125.1		27.8	
PHT	21.3	0.42	-	117.3	42.6	141.1	154.8	36.7	
PHT-b-T-pTHF 50/50	53.5	0.17	−65.9	87.8	21.5	121.1		24.8	
PHN	55.4	0.04	-	178.5	45.3	212.5		53.5	
PHN-b-N-pTHF 50/50	36.1	0.22	−35.8	155.2	23.5	182.6		33.1	

T_gR_, T_gF_ glass transition temperature of the rigid segment from second heating, and glass transition of the flexible segment from second heating, respectively; ∆C_p_, change of heat capacity; T_c_, ∆H_c_, crystallization temperature and the corresponding enthalpy of crystallization; T_m_, ∆H_m_, melting temperature and the corresponding enthalpy of melting.

**Table 5 materials-14-04614-t005:** Properties determined from DMTA analysis for synthesized polyesters and copolymers.

Sample	E’ at 25 °C [MPa]	T_α1_ [°C]	T_α2_ [°C]
PHF	2577	46.7	-
PHF-b-F-pTHF 50/50	181	69.4	−32.4
PHT	2105	43.2	-
PHT-b-T-pTHF 50/50	151	61.8	−40.4
PHN	2387	73.2	-
PHN-b-N-pTHF 50/50	152	43.9	−17.3

E’—storage modulus at 25 °C; T_α1_, T_α2,_ temperatures of α-relaxations (α_1_ and α_2_).

**Table 6 materials-14-04614-t006:** TGA data: temperatures of 5%, 10%, 50%, 90% mass loss, the temperatures corresponding to Table 1 and T_DTG2_) in an oxidizing and inert atmosphere.

Sample	T_5%_ (°C)	T_10%_ (°C)	T_50%_ (°C)	T_90%_ (°C)	T_DTG1_ (°C)	T_DTG2_ (°C)
**Measurement in an Oxidizing Atmosphere**						
PHF	362	369	386	409	386	470
PHF-b-F-pTHF 50/50	328	341	378	434	386	494
PHT	374	380	401	481	400	500
PHT-b-T-pTHF 50/50	348	363	398	501	398	503
PHN	380	388	409	532	406	529
PHN-b-N-pTHF 50/50	354	368	405	527	405	521
**Measurement in an inert atmosphere**						
PHF	362	369	385	405	385	-
PHF-b-F-pTHF 50/50	334	348	385	409	391	-
PHT	370	380	402	471	403	-
PHT-b-T-pTHF 50/50	366	377	404	432	405	-
PHN	382	390	412	-	409	-
PHN-b-N-pTHF 50/50	378	387	411	616	411	-

T_5%_, T_10%_, T_50%_ and T_90%_—temperatures corresponding to 5%, 10%, 50% and 90% of mass loss, T_DTG1_ and T_DTG2_—the temperatures corresponding to the maximum of mass losses.

**Table 7 materials-14-04614-t007:** Tensile properties of synthesized polyesters and copoly(ether-ester)s.

Sample	H [Sh°D]	E [MPa]	σ_y_ [MPa]	ε_y_ [%]	σ_b_ [MPa]	ε_b_ [%]
PHF	65 ± 1	395.8 ± 41.7	23.3 ± 0.4	8.5 ± 0,4	44.8 ± 1.8	319.4 ± 17.3
PHF-b-F-pTHF 50/50	40 ± 1	56.6 ± 7.6	-	-	30.6 ± 1.1	564.9 ± 30.0
PHT	62 ± 1	372.3 ± 33.2	19.6 ± 0.4	13.9 ± 0.6	38.4 ± 3.1	421.4 ± 21.0
PHT-b-T-pTHF 50/50	34 ± 1	57.6 ± 8.9	-	-	15.8 ± 0.2	671.3 ± 15.6
PHN	75 ± 1	1347.3 ± 20.6	41.4 ± 6.0	4.4 ± 1.5	46.0 ± 0.5	340.2 ± 3.1
PHN-b-N-pTHF 50/50	41 ± 1	76.8 ± 3.1	-	-	31.2 ± 0.8	606.9 ± 33.7

E—Young’s modulus (calculated from strain 0.05% to 0.25%); σ_y_—tensile strength at yield; ε_y_—elongation at yield; σ_b_—tensile strength at break; ε_b_—elongation at break, H—hardness.

## Data Availability

The data presented in this study are available on request from the corresponding author. After publication the data will be kept in the open repositories by means of Mendeley Data and RepOD.

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
