# Peer review of "Influence of Rigid Segment Type on Copoly(ether-ester) Properties"

_materials, 2021, doi:10.3390/ma14164614_

Round 1

Reviewer 1 Report

The authors synthesized PHF, PHT, PHN and copolymers (ether-ester)s by two-stage condensation of the polymer melt was
successful. The synthesized materials were investigated and compared in terms of chemical structure, mechanical, thermal, and utilitarian properties. 
The chemical structure of the materials was analyzed by FTIR spectroscopy and 1H quantitative
The materials were studied by the nuclear magnetic resonance (NMR) method. I would like to have a comparative analysis with the literature.   I recommend this article for publication

Reviewer 2 Report

In this work, the author successfully prepared three copoly(ester-ether)s from renewable materials, systematically investigated their thermal and mechanical properties, and discussed their applications as thermoplastic elastomers. This work overall is in good quality and may contribute to engineer thermoplastic elastomers from renewable resources. Questions need be addressed by the author are as below:

  1. One or two sentences should be added to the abstract to describe the key mechanical properties of the synthesized polymers and compare them to commercial TPE. These will give readers a clear notion of the importance of the work. Same suggestion for the conclusion.
  2. The detailed development history of FDCA in the introduction is suggested to be removed because it is not strongly related to the work and adds burden of reading.
  3. In Figure 1 (a) transesterification of polyesters, the number of methanol byproducts should be 2n. Synthetic route for copoly(ether-ester)s is missing in Figure 1(a).
  4. Different heating temperatures were applied to polymers as shown in Table 1, can the author clarify how Tp , Tkt, and Tkp were selected.
  5. In table 2, the title "Temperatures and times used in the synthesis of polyesters and copoly(ether-ester)s." should be changed to "Injection molding procedure".
  6. Which proton correspond to the peak at 8.25ppm in Figure 5 PHN?
  7. In line 386, the author claimed the β-relaxation could be recognized as Tg as is observed, but Tg is usually denoted as Tα. The author needs to double check this.
  8. The subscript for the peak at -17.3 oC for PHN-b-N-pTHF 50/50 in Figure 11 should be 2.
